# Survival trends among people living with human immunodeficiency virus on antiretroviral treatment in two rural districts in Ghana

**Eugene Sackeya**[1,2]*, **Martin Muonibe Beru**[1,3], **Richard Nomo Angmortey**[1,3], **Douglas Aninng Opoku**[4,5], **Kingsley Boakye**[1], **Musah Baatira**[1,6], **Mohammed Sheriff Yakubu**[1,7], **Aliyu Mohammed**[1], **Nana Kwame Ayisi-Boateng**[8], **Daniel Boateng**[1,9], **Emmanuel Kweku Nakua**[1], **Anthony Kweku Edusei**[10]

**1** Department of Epidemiology and Biostatistics, School of Public Health, Kwame Nkrumah University of Science and Technology, Kumasi, Ghana, **2** Tatale District Hospital, Tatale, Northern Region, Ghana, **3** Tamale Teaching Hospital, Tamale Metropolitan, Tamale, Ghana, **4** Allen Clinic, Family Health Services, Kumasi, Ghana, **5** Department of Global Health and Internal Health, School of Public Health, Kwame Nkrumah University of Science and Technology, Kumasi, Ghana, **6** St. Joseph's Midwifery Training College, Jirapa, Ghana, **7** Nursing and Midwifery Training College, Nalerigu-Ghana, **8** Department of Medicine, School of Medicine and Dentistry, Kwame Nkrumah University of Science and Technology, Kumasi, Ghana, **9** Julius Global Health, Julius Centre for Health Sciences and Primary Care, University Medical Centre, Utrecht, The Netherlands, **10** Department of Health Promotion, School of Public Health, Kwame Nkrumah University of Science and Technology, Kumasi, Ghana

* mcsackeya@gmail.com

**Data Availability Statement:** All relevant data are within the manuscript and its Supporting information files.

## Abstract

### Background

The human immunodeficiency virus (HIV) has caused a lot of havoc since the early 1970s, affecting 37.6 million people worldwide. The 90-90-90 treatment policy was adopted in Ghana in 2015 with the overall aim to end new infections by 2030, and to improve the life expectancy of HIV seropositive individuals. With the scale-up of Highly Active Antiretroviral Therapy, the lifespan of People Living with HIV (PLWH) on antiretrovirals (ARVs) is expected to improve. In rural districts in Ghana, little is known about the survival probabilities of PLWH on ARVs. Hence, this study was conducted to estimate the survival trends of PLWH on ARVs.

### Methods

A retrospective evaluation of data gathered across ARV centres within Tatale and Zabzugu districts in Ghana from 2016 to 2020 among PLWH on ARVs. A total of 261 participants were recruited for the study. The data was analyzed using STATA software version 16.0. Lifetable analysis and Kaplan-Meier graph were used to assess the survival probabilities. "Stptime" per 1000 person-years and the competing risk regression were used to evaluate mortality rates and risk.

**Funding:** The authors received no specific funding for this work.

**Competing interests:** The authors have declared that no competing interests exist.

## Results

The cumulative survival probability was 0.8847 (95% CI: 0.8334–0.9209). The overall mortality rate was 51.89 (95% CI: 36.89–72.97) per 1000 person-years. WHO stage III and IV [AHR: 4.25 (95%CI: 1.6–9.71) p = 0.001] as well as age group (50$^+$ years) [AHR: 5.02 (95% CI: 1.78–14.13) p = 0.002] were associated with mortality.

## Conclusion

Survival probabilities were high among the population of PLWH in Tatale and Zabzugu with declining mortality rates. Clinicians should provide critical attention and care to patients at HIV WHO stages III and IV and intensify HIV screening at all entry points since early diagnosis is associated with high survival probabilities.

## Background

The human immunodeficiency virus (HIV), the organism responsible for acquired immune deficiency syndrome (AIDS) has caused a lot of havoc to individuals and families since its outbreak in the year 1981[1]. HIV has affected 37.6 million people worldwide as of 2020, with an estimated 27.4 million people accessing antiretroviral therapy (ART) globally [2].

The majority affected by this HIV pandemic live in low and middle-income countries (LMICs) with 20.6 million (55%) of all cases from eastern and southern Africa, while 13.0% (4.7 million) are in western and central Africa [2]. In Ghana, about 342,307 people are living with HIV (PLWH) and of these, 77% of them are receiving lifesaving highly active ART [3]. The therapeutic value of ART for PLWH is undeniable and early initiation reduces morbidity and mortality [4]. It is evident that when HIV replication is suppressed, patients have significant improvement in immunologic status, as seen by higher clusters of differentiation four (CD4) counts, lower AIDS-related morbidity and death, and, in many cases, a return to a normal or near-normal quality of life [4,5]. The advantages of early ART include a reduction in not just classic AIDS-related problems, but also end-stage organ damage as well as non-AIDS-defining malignancies [4,5].

ART administration was introduced in Ghana in June 2003 as part of a comprehensive care package including voluntary counselling and testing (VCT), prevention of mother-to-child transmission (PMTCT) of HIV, and the treatment and management of sexually transmitted infections (STIs) [6]. However, only PLWH whose CD4 count was ≤ 350 or classified as being in WHO stages III and IV were initiated on antiretrovirals (ARVs) [6]. Owing to the global impact of the HIV pandemic exceeding all expectations, with a mortality peak of 1.9 million deaths in 2006 to 0.95 million in 2017 [7], the World Health Organization (WHO) and its partners adopted an ambitious treatment target, dubbed the 90-90-90 agenda which was reviewed in 2020 to 95-95-95 [8]. In this policy, 95% of PLWHs should know their status, 95% of those who know their status should be on ARVs and 95% of those on ARVs should achieve viral suppression [8]. This treatment policy is in line with sustainable development goal three which emphasizes the need for good health and well-being of all people by 2030 with HIV being one of the significant indicators [9].

Ghana, a WHO member state adopted the policy in 2015 with efforts made to increase accessibility to HIV counselling and testing services, provision of ARVs for all those living with HIV, and enhance evaluation of the effectiveness of ARVs through viral load assessment

at regular intervals at no cost to the clients [3]. In 2019, 58% of PLWH in Ghana knew their HIV status, of this number, 77% are receiving ARVs and 68% of those receiving ARVs had their viral load suppressed [3].

During the early 1980s, to be diagnosed with HIV in Ghana was synonymous with a death sentence, however, with the scale-up of ART, more PLWHs are now receiving the lifesaving ARVs to improve their life expectancy [10]. In Ghana, there is limited data on the survival trends of PLWH. This study therefore sought to estimate the survival trends of PLWH on ARVs for the periods between 2016 to 2020, the mortality rate and the associated risk factors.

## Methods

### Study design

The study was a retrospective evaluation of data from antiretroviral therapy (ART) sites in two rural districts in Ghana (Zabzugu and Tatale Districts).

### Study setting

The study was conducted in the Zabzugu and Tatale districts of the northern region of the Republic of Ghana. These two districts were purposively selected based on their performance at the regional annual reviews for 2019 and 2020 where they were rated among the top five districts in terms of their active testing and search for new cases, the enrollment of all such cases on ART and the quality of their monthly reports in the region. Participants were drawn from four ART centres at the Zabzugu District Hospital (ZDH), Tatale District Hospital (TDH), Kpalbutabu Health Centre -Tatale and Nakpali Health Centre -Zabzugu. The ZDH located in the capital of Zabzugu district was the first to introduce ART service while TDH was marked as a centre located in the district capital two years later. The two health centres located in Zabzugu and Tatale were introduced to reduce the distance of travel for clients who were far from the two centres located at the district capitals.

### Study population

The study population included PLWH, 13 years old and above, in the Tatale and Zabzugu districts who were diagnosed and enrolled on ARVs. The study participants were PLWH who started ARVs in 2016 through to 2020. PLWH less than 13 years old at the time of registration due to their dependency on parents and caregivers likely to influence adherence and those who were transferred from districts outside the two districts after they were initiated on ART were excluded.

### Sampling technique

The entire cohort of PLWH who started ART in the Zabzugu and Tatale Districts from January 1, 2016, to December 31, 2020, with 5 years of follow-up or less based on the year of initiation, were recruited for the study. A total of 261 participants were recruited for this study this constituted the entire number of PLWH who started their treatment from 2016 to 2020 and were eligible for inclusion (Fig 1). A total of 275 PLWH were registered within the period, while 14 participants were excluded from the study as a result of transfers, under age (<13 years) and incomplete information. Most of the participants entered the study at different entry points but 31st December 2021 was the administrative censoring date for all participants.

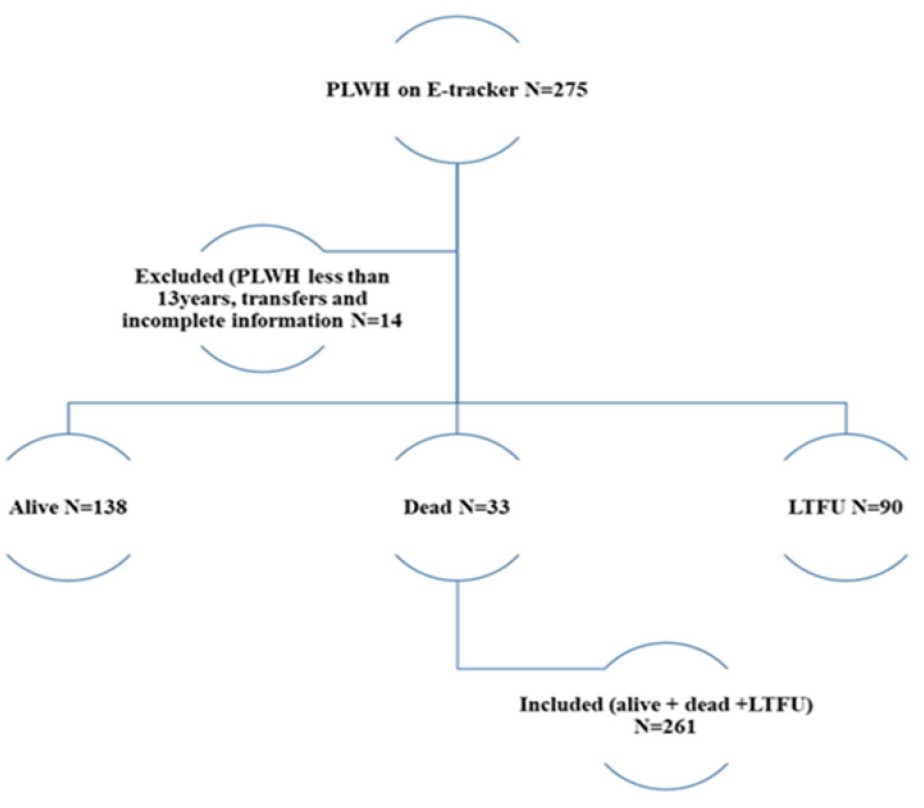

**Fig 1. Inclusion of PLWH for analysis.**

## Variables description

**Dependent variable.** The main outcome variable was the survival probabilities of PLWH on ARVs. This predicted the chance of a participant being alive following the initiation to ARVs during the study periods. The secondary outcomes included the mortality rate and the risk factors for mortality.

**Independent variables.** The study used eleven (11) independent variables. These included the type of regimen the person was on, which is either an efavirenz-based combination or a dolutegravir-based combination, HIV status disclosure refers to disclosing one's HIV status to anyone other than a healthcare provider, and enrolment on the National Health Insurance Scheme (NHIS). The remaining variables included the participant's marital status, WHO HIV stage at the time of diagnosis, client's locality or area of residence, client's health facility where ARVs are received, gender and religion.

## Data collection and management

Data collection was conducted from 1$^{st}$ July 2022 to 31$^{st}$ August 2022 using a standard checklist by two public health nurses and a postgraduate student of Epidemiology and Biostatistics. This was done by reviewing existing medical records including the patient's ARVs folder, the pharmacy logbook, the ARVs logbook, and the viral load register. Two different data collection officers collected the same datasets. After a review of the data, two forms from Data Officer One and five forms from Data Officer Two were incomplete and hence were returned for completion. Fourteen forms were dropped because they did not meet the eligibility criteria. The

data was validated by comparing the data from the two research assistants to ensure accuracy and consistency. After cleaning, the data was stored in a hard drive and also in a Google drive for future use and protection.

## Data analysis

Data was analyzed using the STATA statistical software version 16.0. Percentages, frequencies, median and interquartile range were used to describe participants' socio-demographic and clinical characteristics. For the person-time computation, the date of administrative censorship (December 31, 2021) was considered the last date for all persons who were still alive and receiving treatment. For the assessment of the survival probabilities, lifetable analysis was done alongside the display of a survival function using a Kaplan-Meier graph to assess variations in survival among subgroups. The overall mortality rate was calculated using stptime per 1000 person-years. The stptime command computed and tabulated the person-time and the incidence rate. The subgroup mortality rates were calculated using stptime adjusting for the subgroup per 1000 person-years. To examine the association between socio-demographic characteristics, clinical characteristics and mortality, the Competing risk regression model was used where loss to follow-up was treated as the competing risk to mortality following the initiation of ARVs. Statistical significance was set at p-value <0.05 at a confidence interval of 95%. Variables (both significant and insignificant at the Univariate level) that were deemed to be important based on previous study [11] were included in the multivariable analysis. The results of the univariate level analysis were reported in the sub-distribution hazard ratio (SHR) while the multivariable level analysis was reported in the adjusted sub-distribution hazard ratio (ASHR).

## Ethical consideration

Ethical clearance was granted by the Committee for Human Research, Publication and Ethics of the Kwame Nkrumah University of Science and Technology with reference number CHRPE/AP/287/22. Participants informed consent was not taken since this study involved a review of secondary data from hospital medical records and the patients were not available at the time of the data collection. This was explained in the application for the ethical clearance and participants' consent was waived by the ethics committee. All the study data that were collected were completely anonymized.

## Results

### Socio-demographic characteristics of participants

The study participants who were in the age category of 30 to 39 years were 41.0% with a median age of 35 years (interquartile range, 14 years). More than half (66.28%) of study participants were females while a little over one-third (42.53%) were Christians. Approximately 51.34% resided in the Tatale Sanguli district even though 58.62% received their medications from the Zabzugu district. Those who had no formal education constituted 72.03% of the participants while 61.30% were non-skilled workers. Approximately 89.7% were married and 90.42% of them were registered with the National Health Insurance Scheme (NHIS) (Table 1).

### Clinical characteristics of the study participants

Table 2 presents the results of the clinical characteristics of the study participants. Majority (64.0%) were diagnosed at WHO stage I and 42.91% had disclosed their HIV status to another person other than the healthcare provider. Most of the participants (83.14%) were on

**Table 1. Socio-demographic characteristics of participants.**

| Variables | Zabzugu District Frequency (%) | Tatale District Frequency (%) | Total (%) |
|---|---|---|---|
| Age (years) | | | |
| **13–19** | 2 (0.8) | 10 (3.8) | 12 (4.6) |
| **20–29** | 32 (12.3) | 31 (11.9) | 63 (24.2) |
| **30–39** | 74 (28.4) | 33 (12.6) | 107 (41.0) |
| **40–49** | 28 (10.7) | 14 (5.4) | 42 (16.1) |
| **50$^+$** | 17 (6.5) | 20 (7.6) | 37 (14.1) |
| Median age (IQR) | **35 (14)** | | |
| Gender | | | |
| **Male** | 59 (22.6) | 29 (11.1) | 88 (33.7) |
| **Female** | 94 (36.0) | 79 (30.3) | 173 (66.3) |
| Religion | | | |
| **Christian** | 50 (19.2) | 61 (23.1) | 111 (42.5) |
| **Muslim** | 64 (24.5) | 17 (6.5) | 81 (31.1) |
| **Traditional** | 39 (14.9) | 30 (11.5) | 69 (26.4) |
| District-of-Residences | | | |
| **Zabzugu district** | 118 (45.2) | 1 (0.4) | 119 (45.6) |
| **Tatale district** | 29 (11.1) | 105 (40.2) | 134 (51.3) |
| **Other districts** | 6 (2.3) | 2 (0.8) | 8 (3.1) |
| Educational Level | | | |
| **Formal education** | 36 (13.8) | 37 (14.2) | 73 (28.0) |
| **No formal education** | 117 (44.8) | 71 (27.2) | 188 (72.0) |
| Occupation | | | |
| **Unemployed** | 50 (19.2) | 30 (11.5) | 80 (30.7) |
| **Skilled worker** | 12 (4.6) | 9 (3.5) | 21 (8.1) |
| **None skilled worker** | 91 (34.9) | 69 (26.4) | 160 (61.3) |
| Marital Status | | | |
| **Never Married** | 10 (3.8) | 17 (6.5) | 27 (10.3) |
| **Ever Married** | 143 (54.8) | 91 (34.9) | 234 (89.7) |
| NHIS Status | | | |
| **Registered with NHIS** | 139 (53.3) | 97 (37.1) | 236 (90.4) |
| **Not registered with NHIS** | 14 (5.4) | 11 (4.2) | 25 (9.6) |

NHIS = National Health Insurance, ZDH = Zabzugu District Hospital, TDH = Tatale District Hospital, IQR: Interquartile range.

Efavirenz-based Combination. A little over half (52.87%) were alive and traceable at the time of the study while 12.6% were dead.

### Five-year trend of survival probabilities

A total of 635.981 person-years with 2.17 median person-years of follow-up was contributed by the 261 participants. In the first year of follow-up, 15 mortalities were recorded giving a survival probability of 0.9376 (95% Cl: 0.8987–0.9619) for the period. After three years of follow-up, the cumulative mortality was 32 and a survival probability of 0.8337 (95% CI: 0.7704–0.8809). With a decline in mortalities over the years, the overall mortality at the end of the fifth year was 33 and a survival probability of 0.8049 (95% CI: 0.7145–0.8693) (Table 3).

Overall, the Kaplan-Meier survival curve revealed that the majority of fatalities occurred in the first to third years after starting ARVs, thus 15, 10 & 7 respectively. Even though the overall

**Table 2. Clinical characteristics of study participants.**

| Variable | Zabzugu District Frequency (%) | Tatale District Frequency (%) | Total (%) |
|---|---|---|---|
| **WHO HIV Stage** | | | |
| Stage I | 105 (40.2) | 62 (23.8) | 167 (64.0) |
| Stage II | 15 (5.7) | 25 (9.6) | 40 (15.3) |
| Both stage III & IV | 33 (12.6) | 21 (8.0) | 54 (20.6) |
| **HIV status disclosure** | | | |
| Disclosed | 70 (26.8) | 42 (16.1) | 112 (42.9) |
| Not disclosed | 83 (31.8) | 66 (25.3) | 149 (57.1) |
| **Combined ARV type** | | | |
| EBC | 129 (49.4) | 88 (33.7) | 217 (83.1) |
| DBC | 24 (9.2) | 20 (7.7) | 44 (16.9) |
| **ARV's treatment outcome** | | | |
| Dead | 23 (8.8) | 10 (3.6) | 33 (12.6) |
| Alive | 86 (33.0) | 52 (19.9) | 138 (52.9) |
| LTFU | 44 (16.9) | 46 (17.6) | 90 (34.5) |

WHO = World Health Organization, HIV = Human Immune Virus, ARV's = Antiretrovirals, n = Sample Size, LTFU = Loss to Follow Up, DBC = Dolutegravir Based Combination, EBC = Efavirenz Based Combination.

cumulative survival probability for the five-year study period was 0.8049, those who were diagnosed at the WHO stages I and II had a higher survival probability as shown in Fig 2 and so was for women and those in the age group of 13–29 years. Patients who did not disclose their HIV status had a higher survival probability than those who disclosed their status but this trend changed after two years of follow-up where the survival probabilities of those who disclosed their status improved.

## Five-Year Mortality Rates among PLWH on ARVs

Thirty-three (12.64%) participants died during the study period. The total person-years for the five years was 635.98 person-years with an overall mortality rate of 51.89 (95% CI: 36.89–72.97) per 1000 person-years. Mortality rates were above the overall rate among participants 30 years and above, 55.22 (95% CI: 37.60–81.10) per 1000 person-years and males 68.65 (95% CI: 41.39–113.87) per 1000 person-years. The mortality rate was almost even among those who disclosed their HIV status and those who did not with 53.60 (95% CI: 31.75–90.51) per 1000 person-years and 50.69 (95% CI: 32.34–79.48) per 1000 person-years respectively. WHO

**Table 3. Survival probabilities.**

| Total person-years | | | | 635.981 |
|---|---|---|---|---|
| **Median person-years of follow-up** | | | | **2.17** |
| **Years** | **No at-risk** | **Deaths** | **Iterations** | **Survival Probability (95% CI)** |
| **0–1** | 261 | 15 | 41 | 0.9376 (0.8987–0.9619) |
| **1–2** | 205 | 10 | 56 | 0.8847 (0.8334–0.9209) |
| **2–3** | 139 | 7 | 35 | 0.8337 (0.7704–0.8809) |
| **3–4** | 97 | 0 | 40 | 0.8337 (0.7704–0.8809) |
| **4–5** | 57 | 1 | 56 | 0.8049 (0.7145–0.8693) |

CL = Confidence Interval, NO = Number.

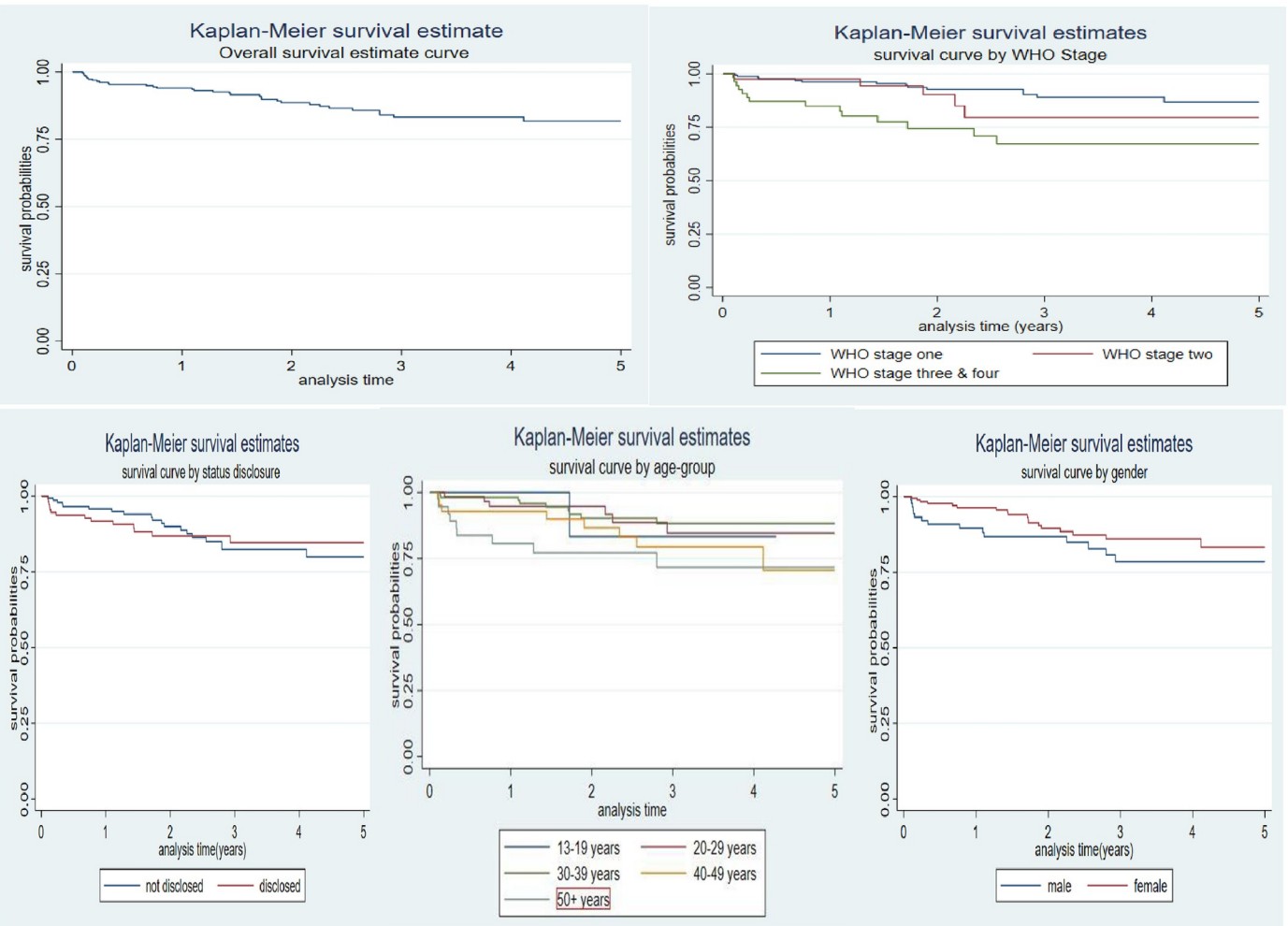

**Fig 2. Kaplan Meier's survival estimates.**

stage III and IV recorded the highest mortality rate of 123.75 (95% CI: 73.29–208.95) per 1000 person-years.

In the univariate competing risk regression analysis, age was statistically significant and associated with mortality. The hazard of mortality was 3.28 times higher among those aged 50 and above [HR: 3.28; 95% CI: 1.29–8.33, p = 0.013] and remained significant after adjusting for confounders in the multivariable analysis [AHR: 5.02; 95% CI: 1.78–14.13, p = 0.002]. WHO stage was statistically significant, the hazard of mortality was 3.51 times higher among those diagnosed at WHO stage III & IV [HR: 3.51; 95% CI: 1.67–7.38, p = 0.001] and the same was the case after adjusting for confounders [AHR: 4.63; 95% CI: 1.91–11.18, p = 0.001] (Table 4).

The hazard of mortality is highest between the first and the second year of follow-up but declined steadily over the rest of the study period from the second year onwards (see Supporting information).

**Table 4. Mortality rates and their determinants.**

| Variable (n = 261) | Dead | PY | Mortality Rate per 1000 PY (95%CI) | Univariate Analysis SHR (95% CI) | P-Value | Multivariable Analysis ASHR (95% CI) | P-Value |
|---|---|---|---|---|---|---|---|
| Overall | 33 | 635.98 | 51.89 (36.89–72.97) | . . . | | . . . | |
| **Age (years)** | | | | | | | |
| 13–19 | 1 | 23.57 | 42.42 (5.98–301.15) | 0.92 (0.12–7.09) | 0.940 | 6.33 (0.78–51.08) | 0.083 |
| 20–29 | 6 | 141.58 | 42.38 (19.04–94.33) | 1.12 (0.40–3.12) | 0.827 | 1.72 (0.54–5.50) | 0.358 |
| 30–39 | 9 | 275.51 | 32.67 (16.10–62.78) | 1 | | 1 | |
| 40–49 | 8 | 111.96 | 71.46 (35.74–142.88) | 2.31 (0.90–5.95) | 0.082 | 1.77 (0.60–5.18) | 0.298 |
| 50$^+$ | 9 | 83.36 | 107.96 (56.17–207.49) | 3.28 (1.29–8.33) | 0.013 | 5.02 (1.78–14.13) | 0.002 |
| **Gender** | | | | | | | |
| Female | 18 | 417.48 | 43.12 (27.16–68.43) | 1 | | 1 | |
| Male | 15 | 218.50 | 68.65 (41.39–113.87) | 1.70 (0. 85–3.37) | 0.132 | 1.06 (0.44–2.51) | 0.893 |
| **Religion** | | | | | | | |
| Traditional | 6 | 185.79 | 32.29 (14.51–71.88) | 1 | | 1 | |
| Christian | 13 | 255.69 | 50.84 (29.52–87.56) | 1.38 (0.52–3.67) | 0.516 | 2.08 (0.73–5.96) | 0.173 |
| Muslim | 14 | 194.50 | 71.98 (42.63–121.53) | 2.20 (0.83–5.79) | 0.111 | 2.74 (0.94–7.99) | 0.065 |
| **Education** | | | | | | | |
| Formal Education | 9 | 150.62 | 59.75 (31.09–114.84) | 1 | | 1 | |
| No formal Education | 24 | 485.36 | 49.45 (33.14–73.77) | 1.02 (0.47–2.19) | 0.963 | 0.58 (0.23–1.45) | 0.247 |
| **Occupation** | | | | | | | |
| Unemployed | 6 | 191.11 | 31.40 (14.10–69.88) | 1 | | 1 | |
| Skilled worker | 3 | 53.49 | 56.08 (18.09–173.89) | 1.96 (0.49–7.79) | 0.337 | 2.41 (0.43–13.42) | 0.315 |
| None Skilled | 24 | 391.38 | 61.32 (41.10–91.49) | 2.06 (0.85–4.99) | 0.111 | 2.23 (0.82–6.07) | 0.117 |
| **Marital status** | | | | | | | |
| Never married | 1 | 59.83 | 16.71 (2.35–118.66) | 1 | | 1 | |
| Married | 32 | 576.15 | 55.54 (39.28–78.54) | 3.91 (0.54–28.19) | 0.177 | 5.16 (0.97–27.50) | 0.054 |
| **NHIS status** | | | | | | | |
| Not Registered | 3 | 59.79 | 50.17 (16.18–155.56) | 1 | | 1 | |
| Registered | 30 | 576.19 | 52.07 (36.40–74.47) | 1.05 (0.31–3.56) | 0.931 | 1.36 (0.40–4.61) | 0.620 |
| **Health facility** | | | | | | | |
| TDH | 10 | 234.80 | 42.59 (22.92–79.15) | 1 | | 1 | |
| ZDH | 23 | 401.18 | 57.33 (38.10–86.27) | 1.66 (0.79–3.48) | 0.181 | 2.05 (0.75–5.60) | 0.161 |
| **Combined ARV type** | | | | | | | |
| DBC | 2 | 77.48 | 25.81 (6.46–103.21) | 1 | | 1 | |
| TBC | 31 | 558.50 | 55.51 (39.04–78.93) | 2.71 (0.64–11.46) | 0.176 | 2.16 (0.38–12.45) | 0.386 |
| **Status disclosure** | | | | | | | |
| Not disclosed | 19 | 374.80 | 50.69 (32.34–79.48) | 1 | | 1 | |
| Disclosed | 14 | 261.18 | 53.60 (31.75–90.51) | 1.06 (0.53–2.11) | 0.861 | 1.20 (0.57–2.53) | 0.628 |
| **WHO stage** | | | | | | | |
| Stage I | 14 | 436.16 | 32.10 (19.01–54.20) | 1 | | 1 | |
| Stage II | 5 | 86.69 | 57.68 (24.01–138.58) | 1.65 (0.60–4.51) | 0.332 | 1.53 (0.50–4.67) | 0.454 |
| Stage III&IV | 14 | 113.13 | 123.75 (73.29–208.95) | 3.51 (1.67–7.38) | 0.001 | 4.63 (1.91–11.18) | 0.001 |

n = Sample Size, PY = Person Years, CI = Confidence Interval, Edu = Education, ZDH = Zabzugu District Hospital, TDH = Tatale District Hospital, TBC = Tenofovir Based Combination, DBC = Dolutegravir Based Combination, WHO = World Health Organization, SHR = Sub-division Hazard Ratio, ASHR = Adjusted Sub-division Hazard Ratio.

## Discussion

Preventing major AIDS-related and non-AIDS-related illnesses in HIV-infected individuals requires early initiation of antiretroviral medication [12]. The majority of the participants were initiated on ARVs at WHO stages I and II (79.31%) and this resonates with earlier reports that most PLWH in Ghana are initiated on ARVs at WHO stages I to III [13,14] This was however different from what was reported in an expanded study involving most of the HIV sentinel sites in Ghana where only 27.7% of PLWH were initiated at WHO stage I and II [15]. The variation in findings could be because the study was conducted during the period of the 90-90-90 treatment policy where there was a more accelerated action for HIV testing and treatment. During that period, testing was done at the OPD instead of the laboratory where only clients who were suspected of being infected were referred for testing. Providing testing services for all persons, whether symptomatic or not, meant that more clients could be diagnosed in the earlier stages of the disease.

The current study identified the majority of PLWH were alive and on treatment at the end of the five years of follow-up. Our findings were a little different from the findings in a study in a primary public hospital of Wukro, Tigray, Ethiopia, where 11% of LTFU was recorded [16]. The reason for the high rate of LTFU could be due to the high numbers diagnosed at WHO stages I and II, who are asymptomatic and may not see the need to take their medications. The overall survival probabilities for the first and fifth years were 0.9376 and 0.8049, respectively. These findings, projected high survival probabilities when compared with another study conducted in the Lawra and Jirapa districts in the Upper West Region of Ghana where the survival probability at the end of the three years was 0.795 per 1000 person-years [17]. The findings were similar to the findings of a study conducted in the Henan province of China from 2005 to 2014 where the cumulated survival for the first and fifth year were 93.7% and 85.3% respectively [18]. The reason for the higher survival probabilities in this study could be due to the removal of all barriers to the initiation of ARVs and early diagnosis due to the 90-90-90 policy which was not the case during the period of the study by Okyere *et al.* [17].

Significant differences in survival probabilities were observed among specific variables using the Kaplan-Meier survival curve. WHO stages I and II had the highest survival probability and this finding agreed with studies in Ghana's Upper West Region and Oromiyaa, Ethiopia where WHO stages I and II had the highest survival probability than stages III and IV [17,19]. The prognosis of the disease at WHO stages I and II at the time of diagnosis and subsequent initiation of the ARVs early could account for the above findings. Although disclosure of one's status may play a role in controlling the spread of HIV, it had very little impact on the improvement of the survival probability of the clients in the short term. However, in the long term, status disclosure could give an individual a better chance of survival compared to someone who has not disclosed the status. The reason for these dynamics could be due to the support that those who disclose their status receive from their treatment supporters. For instance, a study in Ghana reported that HIV status disclosure is significant for having a functional family [20] and family functionality among PLWH also improves treatment outcomes (viral suppression) [14] which could enhance survival probabilities. Females had better survival probabilities than their male counterparts and this could be attributed to the health-seeking behaviors of females [21].

The overall mortality rate observed was low compared with what was found in a retrospective cohort study from 2004–2013 in Nepal [22]. The reason for the reduction in overall mortality in this study may be because the majority of the participants were diagnosed at WHO stages I and II compared to 16.2% which was reported in the Nepal study [22]. Mortality was highest among those diagnosed at the WHO stages III and IV, meaning this group will need

special attention, especially within the first two years of diagnosis when most of the deaths occurred. Regular clinic visits, viral load monitoring, and adherence to the ARV regimen are essential to prevent mortalities. Other areas where significant variations in mortality were observed were gender and religion. Morality was high among males compared to females and that could be attributed to the health-seeking behaviors of the male gender [21].

The hazard of mortality after the initiation of ARVs was statistically significant and associated with those in WHO stages III and IV and age group 50$^+$ years and after adjusting in a multivariable analysis, WHO stages III and IV and age group 50$^+$ years remained significant. The findings were similar to other studies that were conducted in Ghana, sub-Saharan Africa, and other parts of the world [18,19,22,23]. From the smooth hazard curve, the risk of mortality was highest in the first and second years of ARV initiation. More pragmatic efforts including regular clinic visits, early diagnosis of opportunistic infections, viral load monitoring and so on, are required to reduce the death rate within the first and second years of ARV initiation.

## Strengths and limitations

This is the first study of its kind on this subject in the Zabzugu and Tatale districts and will contribute greatly to the literature and also serve as a baseline finding for future studies in the two districts. The lack of a comparison group restricted our ability to compare the outcome between those initiated on ARVs and those that are not on ARVs. Hence, a comparison was done using previous studies while the presence of missing data also restricted the inclusion of variables such as viral load. Comorbidity and coinfection were also excluded due to the lack of adequate information on such an important variable and that has also limited our analysis.

## Conclusion

Survival probabilities were high among PLWH in the two districts compared to previous studies in other districts conducted before the introduction of the treat-all policy. We recommend the need for early diagnosis through testing at all entry points to hospitals and clinics and through the "Know your Status" campaigns for HIV. Clients who are 50 years and above and those who were diagnosed at the WHO stages III and IV should be given shorter clinic visit intervals and ensure viral load monitoring is done as recommended by the National AIDS Control Program.

## Supporting information

**S1 File. Mortality hazard curve.**
(TIF)

**S1 Data. HIV survitrend data (1) (datasets).**
(XLSX)

## Acknowledgments

We are grateful to all PLHIV whose records we reviewed for this study. To all the staff of the four ARV centres in both Tatale and Zabzugu, we thank them for their efforts in saving lives and the documentation without which this study would not have been possible. We are also grateful to the management of the two hospitals and the District Directors of Health Services for their approval of the study. We are most grateful to God for the inspiration, guidance and protection throughout the study.

## Author Contributions

**Conceptualization:** Eugene Sackeya, Martin Muonibe Beru, Richard Nomo Angmortey, Douglas Aninng Opoku, Kingsley Boakye, Musah Baatira, Mohammed Sheriff Yakubu, Aliyu Mohammed, Daniel Boateng, Emmanuel Kweku Nakua, Anthony Kweku Edusei.

**Data curation:** Eugene Sackeya, Martin Muonibe Beru, Richard Nomo Angmortey, Musah Baatira.

**Formal analysis:** Eugene Sackeya, Martin Muonibe Beru, Richard Nomo Angmortey, Kingsley Boakye, Musah Baatira, Mohammed Sheriff Yakubu.

**Methodology:** Eugene Sackeya, Martin Muonibe Beru, Richard Nomo Angmortey, Douglas Aninng Opoku, Kingsley Boakye, Musah Baatira, Mohammed Sheriff Yakubu, Aliyu Mohammed, Daniel Boateng, Emmanuel Kweku Nakua, Anthony Kweku Edusei.

**Project administration:** Eugene Sackeya, Martin Muonibe Beru, Richard Nomo Angmortey, Kingsley Boakye, Mohammed Sheriff Yakubu.

**Supervision:** Aliyu Mohammed, Daniel Boateng, Emmanuel Kweku Nakua, Anthony Kweku Edusei.

**Validation:** Douglas Aninng Opoku, Aliyu Mohammed, Daniel Boateng, Emmanuel Kweku Nakua, Anthony Kweku Edusei.

**Writing – original draft:** Eugene Sackeya, Martin Muonibe Beru, Richard Nomo Angmortey, Kingsley Boakye, Musah Baatira, Mohammed Sheriff Yakubu.

**Writing – review & editing:** Douglas Aninng Opoku, Aliyu Mohammed, Nana Kwame Ayisi-Boateng, Daniel Boateng, Emmanuel Kweku Nakua, Anthony Kweku Edusei.

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
