## [Decision Letter · Decision Letter 0]

10 Nov 2023

PONE-D-23-25825Survival trends among people living with human immunodeficiency virus on antiretroviral treatment in two rural districts in GhanaPLOS ONE

Dear Dr. Sackeya,

Thank you for submitting your manuscript to PLOS ONE. After careful consideration, we feel that it has merit but does not fully meet PLOS ONE’s publication criteria as it currently stands. Therefore, we invite you to submit a revised version of the manuscript that addresses the points raised during the review process.

We look forward to receiving your revised manuscript.

Kind regards,

Billy Morara Tsima, MD MSc

Academic Editor

PLOS ONE

Journal Requirements:

Reviewers' comments:

Reviewer's Responses to Questions

**Comments to the Author**

1. Is the manuscript technically sound, and do the data support the conclusions?

Reviewer #1: Partly

Reviewer #2: Yes

2. Has the statistical analysis been performed appropriately and rigorously? 

Reviewer #1: I Don't Know

Reviewer #2: Yes

3. Have the authors made all data underlying the findings in their manuscript fully available?

Reviewer #1: Yes

Reviewer #2: Yes

4. Is the manuscript presented in an intelligible fashion and written in standard English?

Reviewer #1: No

Reviewer #2: Yes

5. Review Comments to the Author

Reviewer #1: There are some language errors in the text, mainly that at times it is unclear which study the author is referring to. I have made some language suggestions in the attached document. The referencing should be reviewed, there are more recent and widely accepted references for early ART initiation, some of the references are not the original source. Kindly review and cite the original sources.

The data presented does not really support the conclusions. Whereas there is a lot of evidence that early ART initiation enhances survival outcomes it is not shown in this study as there is no comparator. Perhaps a comparison of survival before and after Treat all would have been more appropriate and would support the authors assertion that " survival probabilities have improved significantly among PLWH in the two districts". Currently the study reads more like a description of survival of PLWHI who are on ART.

The setting description may be better with a description of HIV services, general mortality rates etc of the district. It is unclear to me why the two districts were selected, what does "active involvement in ART mean"?

The authors say they used competing risk analysis, I believe they should state in the methods what the competing risk was( or the typeof competing risk) and that they used subdivision hazard ratios. One of the independent variables mentioned in the methods- comorbidity/coinfection is not accounted for in the results.

The results should also account for and explain the excluded samples. This is an important independent variable as it could include HIV define illnesses )WHO stage 3 or 4) or non-communicable diseases with cardiovascular complications e.g. hypertension and Diabetes which may cause competing causes of death.

The rest of my comments are shown in the attachment.

Reviewer #2: The manuscript was very detailed, easy to comprehend and results were explained in a lucid manner. The authors did a good job in understanding the importance of the topic and presenting it in a precise manner.

6. PLOS authors have the option to publish the peer review history of their article (what does this mean?). If published, this will include your full peer review and any attached files.

Reviewer #1: No

Reviewer #2: No

---

## [Author Response · Author response to Decision Letter 0]

24 Dec 2023

Academic Editor

Comments: Please review your reference list to ensure that it is complete and correct. If you have cited papers that have been retracted, please include the rationale for doing so in the manuscript text, or remove these references and replace them with relevant current references. Any changes to the reference list should be mentioned in the rebuttal letter that accompanies your revised manuscript. If you need to cite a retracted article, indicate the article’s retracted status in the References list and also include a citation and full reference for the retraction notice.

Response: We appreciate your valuable comments. These changes have been made in the revised manuscript.

List of references removed from the revised draft

zolopa AR. The evolution of HIV treatment guidelines: Current state-of-the-art of ART. Antiviral Res. 2010 Jan;85(1):241–4.

Owusu-ansah Y. Socio-economic effect of HIV / AIDS in Ghana. 2009;(December 2002). Available from: https://www.ghanaweb.com/GhanaHomePage/features/Socio-economic effect-of-HIV-AIDS-in-Ghana-3098

Center for Disease Control and Prevention. HIV Basics | HIV/AIDS | CDC [Internet]. 2022 [cited 2022 Apr 6]. Available from: https://www.cdc.gov/hiv/basics/index.html

Ghana Statistical Service. Zabzugu district. 2010. 1–10 p. 

Ghana Statistical Service. Tatale sanguli district. 2010. 1–10 p.

List of references currently included that were not part of the original draft. 

AIDS.gov. Global HIV/AIDS Timeline [Internet]. KFF. [cited 2023 Nov 20]. Available from: https://www.kff.org/global-health-policy/timeline/global-hivaids-timeline/

Lopez-Cortes LF, Gutiérrez-Valencia A, Ben-Marzouk-Hidalgo OJ, Hospital Universitario Virgen del Rocio, Seville, Spain. Antiretroviral Therapy in Early HIV Infection. N Engl J Med. 2016 Jan 28;374(4):393–4. 

TEMPRANO ANRS 12136 Study Group, Danel C, Moh R, Gabillard D, Badje A, Le Carrou J, et al. A Trial of Early Antiretrovirals and Isoniazid Preventive Therapy in Africa. N Engl J Med. 2015 Aug 27;373(9):808–22.

Ayisi-Boateng NK, Enimil A, Essuman A, Lawson H, Mohammed A, Aninng DO, et al. Family APGAR and treatment outcomes among HIV patients at two ART Centres in Kumasi, Ghana. Ghana Med J. 2022 Sep;56(3):160–8.

Ayisi-Boateng NK, Enimil A, Mohammed A, Essuman A, Lawson H, Opoku Aninng D, et al. Predictors of family functionality amongst human immunodeficiency virus-serodiscordant couples in two major hospitals in Kumasi, Ghana. Afr J Prim Health Care Fam Med. 2020 Jun 9;12(1):e1–6.

Reviewer 1

Comments: There are some language errors in the text, mainly that at times it is unclear which study the author is referring to. I have made some language suggestions in the attached document.

Response: The language errors have been addressed in the revised manuscript.

Comments: The use of the abbreviation HAART and ART

Response: Your concern of the use of the terms HAART and ART concurrently have been addressed. Whiles the meaning as used in this work is relatively different, to avoid creating confusion among readers, we have dropped the abbreviation HAART and will maintain the abbreviation ART throughout this manuscript.

Comment: A more relevant setting description would relate to the provision of HIV care in these districts

Response: This has been duly revised to reflect the provision of care to PLWH in the two districts 

“These two districts were purposively selected based on their performance at the regional annual reviews for 2019 and 2020 where they were rated among the top five districts in terms of their active testing and search for new cases, the enrollement of all such cases on ART and the quality of their monthly reports.. Participants were drawn from four ART centers at the Zabzugu district hospital (ZDH), Tatale district hospital (TDH), Kpalbutabu Health Center -Tatale and Nakpali Health Center -Zabzugu. The Zabzugu district hospital located in the capital was the first to introduce ART service while Tatale District Hospital was marked as a center located at the district capital two years later. The two health centers located in Zabzugu and Tatale were introduced to reduce the distance of travel for clients who were far from the two centers located at the district capitals”

Comments: It is unclear what the competing risk in this study were. Since the outcome of interest is death. This is not my area of expertise but it seems KM estimates are not recommended if considering competing risks, cumulative incidence may be better. In the results the auther does show SHR in one of the tables but it should be mentioned here as well

Response: The competing risk has been clearly stated following the request, we think it was an oversight. 

KM estimates targets objective one where we are looking at the survival probabilities. It serves as sub analysis providing information on survival probabilities among sub-groups.

 Competing risk on the other hand targets a different objective where we are trying to estimate the mortality hazards among PLWH on ARVs. Because the model can provide hazard ratios taking into consideration the people who are lost to follow-up, a property that other models such as cocks proportional hazard lacks, made it suitable for consideration looking at the number of people who were lost to follow-up in this study. This makes the KM estimates and the competing risk regression independent of each other and we think it’s possible to maintain both in the manuscript. 

The SHR as seen in the results has also been duly captured in the methodology as suggested.

Comments: Not clear why 13years old are the cut-off

Response: In the era of treat all, PLWH do not need a treatment supporter. This is not the case with children as they continue to rely on their parents for their medication as well as their support to take them. In the study setting, child care or upbringing is a whole family affair and mothers will do anything to conceal their HIV status including failure to go for ARVs for the children. Already child mortalities are high in the study setting caused by many factors including anemia associated with malaria, malnutrition, pneumonia, etc. For a survival analysis which is, retrospective in nature, it will be difficult to control all the other factors of mortality and that is why we failed to report on the co-morbidity and co-infection in the results as you rightly mentioned in your review. These are the reasons for which the authors restricted the analysis to 13years and above.

Comment: Please describe- typical census means all participants within a defined area are included (complete enumeration) therefore it is not sampling

Response: The use of census was in context to mean all PLWH initiated on ARVs between 1st January 2016 to 31st December 2020 however, to have used census and sampling at the same time appears contradictory hence we have revised that section to read,

“The entire cohort of PLWH who started ART in the Zabzugu and Tatale Districts from January 1, 2016, to December 31, 2020, with 5 years of follow-up or less based on the year of initiation, were recruited for the study”.

Comment: Comorbidity/coinfection not accounted for in the results.

Response: This is true and we have reviewed the methodology to reflect only what was accounted for in the results section. 

Comment: This should be accounted for, how many were returned to be completed, what was missing and how many were dropped, it may also be helpful to di a sub analysis, “Two different data collection officers collected the same data set and incomplete forms were returned to be completed and those that were not completed for lack of adequate information were dropped”.

Response: This has been accounted for as requested. 

“Two different data collection officers collected the same data set. Two incomplete forms from data officer one and five forms from data officer two were returned to be completed. Fourteen forms were dropped because they did not meet the eligibility criteria.”

Comments: This should be clearer or more precise.

“. Descriptive statistics were used to describe the baseline characteristics of participants including the socio-demographic and the clinical characteristics”

Response: This has also been revised to read, 

“Percentages, frequencies, median and interquartile range were used to describe the participants socio-demographic and the clinical characteristics” 

Comments: Flow chart of final sample size

Response: This has been duly provided

Comment: Generally, this is not used for formal inference. I would recommend using it as an appendix

Response: The recommendation well acknowledged and estimated curve of mortality hazard figure moved to the appendix.

Comments: The referencing should be reviewed, there are more recent and widely accepted references for early ART initiation, some of the references are not the original source. Kindly review and cite the original sources.

Response: All the references have been addressed in the revised manuscript as suggested. The START and Temprano studies the reviewer recommended have also been useful as they strengthen the evidence of the current study. 

Comments: The data presented does not really support the conclusions. Whereas there is a lot of evidence that early ART initiation enhances survival outcomes it is not shown in this study as there is no comparator. Perhaps a comparison of survival before and after Treat all would have been more appropriate and would support the author’s assertion that “survival probabilities have improved significantly among PLWH in the two districts". Currently the study reads more like a description of survival of PLWHI who are on ART.

Response: The use of a comparator would have enhanced the study’s findings. All PLWH were enrolled on ART due to the change in the treatment policy and the study design made it difficult to identify those who were not on ART to serve as a comparator. What the authors did to arrive at the conclusion that there was improvement in survival was to compare the findings with similar studies that were conducted before the treat-all era. The reviewer assertion that instead of improvement of survival we should use high survival rate is well appreciated as this will make it more understanding. We have therefore revised it accordingly.

“Survival probabilities were high among PLWH in the Zabzugu and Tatale districts compared to previous studies in other districts conducted before the introduction of the treat-all policy”.

Comments: The authors say they used competing risk analysis, I believe they should state in the methods what the competing risk was (or the type of competing risk) and that they used subdivision hazard ratios. One of the independent variables mentioned in the methods- comorbidity/coinfection is not accounted for in the results

Response: This has also been addressed as recommended

 ”To examine the association between socio-demographics, clinical characteristics and mortality, the Competing risk regression model was used where loss to follow-up was treated as the competing risk to mortality following the initiation of ARVs”.

Comments: The results should also account for and explain the excluded samples. This is an important independent variable as it could include HIV define illnesses) WHO stage 3 or 4) or non-communicable diseases with cardiovascular complications e.g. hypertension and Diabetes which may cause competing causes of death.

Response: Due to the retrospective nature, data on this sensitive variable was sparse, limiting our ability to do sub-analysis. In light of the reviewer’s assessment, we have revised the methodology to address the point on comorbidity and coinfection where variables that are not captured in the results are not included in the methods section.

Reviewer 2

Comments: The manuscript was very detailed, easy to comprehend and results were explained in a lucid manner. The authors did a good job in understanding the importance of the topic and presenting it in a precise manner.

Response: Thank you for these comments.

---

## [Editor Report · Decision Letter 1]

9 Jan 2024

Survival trends among people living with human immunodeficiency virus on antiretroviral treatment in two rural districts in Ghana

PONE-D-23-25825R1

Dear Dr. Sackeya,

We’re pleased to inform you that your manuscript has been judged scientifically suitable for publication and will be formally accepted for publication once it meets all outstanding technical requirements.

Kind regards,

Billy Morara Tsima, MD MSc

Academic Editor

PLOS ONE
---

## [Editor Report · Acceptance letter]

26 Feb 2024

PONE-D-23-25825R1 

PLOS ONE

Dear Dr. Sackeya, 

I'm pleased to inform you that your manuscript has been deemed suitable for publication in PLOS ONE. Congratulations! Your manuscript is now being handed over to our production team.

Kind regards, 

on behalf of

Dr. Billy Morara Tsima 

Academic Editor

PLOS ONE